# Dynamic Clustering and Scaling Behavior of Active Particles under Confinement

**DOI:** 10.3390/nano14020144

**Published:** 2024-01-09

**Authors:** Matthew Becton, Jixin Hou, Yiping Zhao, Xianqiao Wang

**Affiliations:** 1School of ECAM, College of Engineering, University of Georgia, Athens, GA 30602, USA; becton@uga.edu (M.B.); jixin.hou@uga.edu (J.H.); 2Department of Physics and Astronomy, University of Georgia, Athens, GA 30602, USA; zhaoy@uga.edu

**Keywords:** active particles, simulation, aggregation, confined environment

## Abstract

A systematic investigation of the dynamic clustering behavior of active particles under confinement, including the effects of both particle density and active driving force, is presented based on a hybrid coarse-grained molecular dynamics simulation. First, a series of scaling laws are derived with power relationships for the dynamic clustering time as a function of both particle density and active driving force. Notably, the average number of clusters $\bar{N}$ assembled from active particles in the simulation system exhibits a scaling relationship with clustering time t described by $\bar{N}\propto t^{-m}$. Simultaneously, the scaling behavior of the average cluster size $\bar{S}$ is characterized by $\bar{S}\propto t^{m}$. Our findings reveal the presence of up to four distinct dynamic regions concerning clustering over time, with transitions contingent upon the particle density within the system. Furthermore, as the active driving force increases, the aggregation behavior also accelerates, while an increase in density of active particles induces alterations in the dynamic procession of the system.

## 1. Introduction

Active particles, distinguished by their autonomous operations at the nano- and microscales within fluids, can be differentiated from particles which passively react to their environment. Their unique ability to self-propel and impart energy to a system has recently garnered considerable attention. The applications of active particles span a diverse array, encompassing catalysis [1], toxin detection [2,3], cellular marking [4], wastewater treatment [5,6,7,8,9,10,11,12], CO_2_ scrubbing [13], chemical and biological warfare agent neutralization [14,15,16,17], on-the-fly hydrogel polymerization [18], micropatterning [19], and even energy generation [20]. Indeed, the lion’s share of interest in active particles has been concentrated in the realm of medicine and chemistry, particularly in drug delivery and synthesis [21,22,23,24,25,26,27,28,29,30,31,32,33,34,35,36]. Magnetically controlled active particles, such as those made of iron oxide, have shown great promise in drug delivery as well as in the removal of blockages from blood vessels within the body [37,38,39]. In applications where active particles are abundantly present, such as in microfluidics devices or the human body, systems can be considered confined fluid systems [40]. However, depending on factors like driving force and particle density, active particles within the body may exhibit tendencies to aggregate and cluster together, which could raise potential risks yet to be investigated in biomedical applications [21,41]. One notable feature of active particles is their enhanced diffusion within a fluid system compared to similar passive particles. Moreover, an increase in active particle density corelates with a greater effective diffusion coefficient [42]. Active colloids operate by dissipating energy at the level of the active particles in order to cause motion with no equilibrium equivalent, resulting in clustering under low dissipation and collective motion under high dissipation [43]. The coexistence of active and passive colloidal particles has been shown to prompt clustering and phase separation, forming chains or clusters [44]. The collective behavior of active particles introduces numerous effects not observed in the behavior of individual particles, surpassing the simple summation of individual behaviors due to interactions arising between active particles [45,46,47,48,49,50]. The tendency of colloids to aggregate can likewise demonstrate complexity when considering the autonomous motion of active particles.

Recent research has demonstrated that the directed assembly of colloidal dispersions can be used to synthesize a diverse range of functional materials in a controlled fashion [51,52,53]. While self-assembly is inherent to passive colloids, systems composed of active particles exhibit an augmented capacity for self-assembly, due to the driven nature of these particles [54,55]. This type of active aggregation has been shown to create dynamic clusters that evolve over time, propelled by both fluid velocity and the individual motion of active particles. This mechanism effectively overcomes the long-range repulsion commonly observed in colloidal systems [56,57]. The aggregation of active colloidal particles has been shown to occur naturally in systems with colloidal volume fractions ranging from 3% to 50% [58]. The complexity of active particle aggregation is noteworthy, with behavior resembling living cell aggregates emerging as active colloids self-organize into clusters [59]. Motility-induced clustering and phase separation have been demonstrated in quasi-2D environments [60]. Under weak confinement, autophoretic active particles exhibit both clustering and dispersion modes, dependent on the driving force and interparticle interactions [61]. In many cases, the presence of geometrical confinement grants far different dynamics to active particles as compared to the more widely modeled unbounded cases, highlighting the importance of considering explicit confinement when modeling non-bulk systems [62,63]. The presence of physical or other manner of confinement has a profound effect on the behavior of active particles, as the existence of a boundary introduces complexity to the motion of the driven motors, leading to phase separation, boundary clustering, and other emergent behaviors [64,65,66,67]. Confinement also serve as a tool for predicting and controlling the behavior of active particles such as micromotors [68]. Active colloidal particle aggregation has been shown in previous works in the literature to be heavily dependent on the size and surface properties of the particles [69,70]. In addition, the application of external electric or magnetic fields can also drive the aggregation process [51,71]. However, there remains a lack of research towards the confined collective behavior of active colloids, specifically covering a range of volume fractions and driving forces, especially as it relates to their aggregation [72,73,74]. Consequently, the micrometer-scale behavior of active particles, especially within confined systems, warrants comprehensive investigation, in order to supplement the rapidly growing literature on computational methods for modeling active matter [75].

The focus of this work is to investigate the effects of the density and driving force of active colloidal particles on their clustering behavior within a confined fluid system. To achieve this, a hybrid coarse-grained molecular dynamics model is employed to simulate the clustering and aggregation dynamics of micrometer-sized colloidal particles. Given the crucial role of hydrodynamics in the clustering process, an explicit representation of the fluid is incorporated into the model. Stochastic rotation dynamics, also known as multiparticle collision dynamics, is used to well replicate both long- and short-range hydrodynamic behaviors at a low Reynolds number. The key variables of interest are the particle coverage ratio θ and the driving force F. Here, θ denotes the 2D projection area coverage ratio of the colloidal particles to the 2D size of the simulation box, while F indicates the driving force acting on the colloidal particles along their axial direction. Results reveal that both coverage and driving force have strong effects on the aggregation behavior of the colloidal particles. A set of empirical power laws were derived to describe the rate of cluster formation over time, in which the scaling exponents strongly depend on the particle coverage ratio and driving force applied on each particle. This comprehensive investigation sheds light on the nuanced interplay between particle density, driving force, and clustering behavior, providing valuable insights into the underlying dynamics of active colloidal systems within confined fluid environments.

## 2. Methods and Models

The system of interest involves passive or active particles at the micrometer scale, which can be termed as colloids [76,77,78]. In these colloidal systems, the suspended particles may interact through hydrodynamic, interparticle, and Brownian (or thermal) forces [79]. Our system directly modeled hydrodynamic and Brownian forces through stochastic rotation dynamics, and the interparticle interactions were modeled with the Derjaguin–Landau–Verwey–Overbeek (DLVO) interaction. A multiparticle collision-based method was used to model the solvent. An extended explanation on the choice of DLVO theory to model this system can be found in the Appendix A.

### 2.1. SRD—Explicit Hydrodynamic Solvent

There is a tremendous difference in scale for both length and kinematic time between colloidal particles and solvent beads. For example, a typical colloidal particle (1 μm diameter) occupies a volume comparable to that of 10^3^ water molecules. Thus, coarse graining of the solvent is necessary in order to reasonably reach significant scales of time and size for the simulated system. In this work, we used stochastic rotation dynamics (SRD), also called multiparticle collision dynamics, to establish a computationally efficient and hydrodynamically accurate fluid system capable of handling thermal noise [80]. The coarse-graining length scale was chosen to be smaller than that of mesoscopic colloidal particles but much larger than the natural length scales of the solvent molecules (in this case, water). By adhering to local momentum conservation, the SRD method reproduces Navier–Stokes hydrodynamics at larger length scales. A detailed rationale for choosing SRD in this work is given in the Appendix A.

In the framework of SRD, the solvent is represented by a large number of point-like beads, each possessing a mass of $1\;m_{s}$ (an explanation of units is given below). These beads are termed as the fluid beads, and it is crucial to note that they are not merely composite particles or clusters composed of aggregates of water molecules. Instead, these beads can be considered as a convenient computational tool to facilitate the coarse graining of fluid properties. The SRD works as follows: first is the streaming step, during which the positions and velocities of the fluid beads are calculated directly by integrating Newton’s equations of motion:(1)$$x_{i}\to x_{i}+v_{i}\cdot t_{i}.$$

The forces acting on the fluid beads are produced through collisions with the system walls or active colloidal particles. Notably, there are no direct forces considered between fluid beads, and the simplification of such pairwise calculations significantly contributes to the overall efficiency of the SRD method. Moving to the second step, the collision step simulates the collisions between fluid beads. The system is partitioned into cubic cells, each having a length of 1 σ. During this step, the velocities of all fluid beads are rotated relative to the center of mass velocity (*V*) of their respective cell, as given below:(2)$$v_{i}\to V+\hat{\omega }\left[v_{i}-V\right].$$

During this step, momentum is effectively transferred between the fluid beads, and the rotation procedure can, thus, be viewed as a coarse graining of particle collisions across both time and space. As a result of the local conservation of mass, momentum, and energy, accurate Navier–Stokes hydrodynamic effects can be captured, including those stemming from thermal noise. The SRD method’s coarse graining of the fluid allows for easy control over viscosity and coupling properties, enabling the depiction of phase segregation and reactive hydrodynamics arising from complex solutes. As a validation study to prove the suitability of SRD for studying the behavior of the fluid beads under a confinement environment, Appendix A illustrates the inverse relationship between velocity and size in Stokes flow for spherical beads of different radii traveling through the SRD beads, with the simulation results aligning well with theory. Appendix A shows the mean squared displacement curves for beads subjected to different applied forces, along with their trajectories, showcasing the enhanced diffusion resulting from the driving force acting on the active particles. Notably, the absence of an applied driving force is equivalent to the Brownian motion observed in passive colloidal particles. All simulations were conducted using the LAMMPS software (2 August 2023 version) [81].

### 2.2. Coarse-Grained Molecular Dynamics (CGMD) Active Particles and DLVO Potential

The system of interest in this paper involves a group of attractive colloidal active particles immersed in a fluid under a quasi-2D confinement scheme, as shown in Figure 1. In this configuration, the active particles (schematically depicted as gray beads in Figure 1a) are subjected to various influences, including a driving force resulting from external fields (such as a magnetic field or electric field), DLVO interactions originating from neighboring active colloids, drag force due to hydrodynamic effects, and random forces arising from the thermal fluctuations within the system. In the simulation, colloidal spheres with a mass M are propagated through the velocity Verlet algorithm using a timestep of 0.01 τ. These colloids are immersed in the fluid and interact with the fluid beads through a repulsive-only Weeks–Chandler–Andersen potential (van der Waals force). While the fluid–fluid interactions are coarse-grained using SRD, the colloid–colloid and colloid–fluid interactions are integrated using a normal molecular dynamics procedure.

The CGMD colloid–colloid interaction potential is described by the DLVO potential specifically developed for colloidal interactions [82]:(3)$$U=U_{vdW}+U_{Coul}+U_{Hertz}$$

Here, the total potential energy between two colloidal particles is the summation of the van der Waals (attractive) force, Coulombic (screened electrostatic) force, and Hertz (hard repulsive contact) force. The interparticle colloidal forces are given as:(4)$$U_{vdW}=-\frac{A_{H}}{12}\left[\frac{d^{2}}{r^{2}-d^{2}}+\frac{d^{2}}{r^{2}}+2ln\left(\frac{r^{2}-d^{2}}{r^{2}}\right)\right],$$
(5)$$U_{Coul}=\pi \epsilon_{r}\epsilon_{0}{\left[\frac{4k_{B}T}{q}tanh\left(\frac{q\Psi_{0}}{4k_{B}T}\right)\right]}^{2}\frac{d^{2}}{r}e^{-\kappa \left(r-d\right)}\;,$$
where, for the van der Waals interaction, $A_{H}$ is the Hamaker constant, d is the colloidal particle diameter (1 μm), and r is the distance between the colloidal particle centers. For the Coulombic interaction, $\epsilon_{r}$ is the electric permittivity of the fluid, $\epsilon_{0}$ is the vacuum permittivity (8.854 × 10^−12^ F/m), q is the colloidal particle charge, $\Psi_{0}$ denotes the effective surface potential of the colloid, and κ is the inverse Debye screening length. To model steric repulsion between colloidal particles and prevent overlap or penetration of the particles, a Hertz contact force is included:(6)$$U_{Hertz}=K{\left(d-r\right)}^{\frac{5}{2}}\;,\;r<d$$

The SRD fluid beads interact with the CGMD particles via the Weeks–Chandler–Anderson potential:(7)$$U_{WCA}=\{\begin{array}{c} 4\epsilon \left({\left(\frac{\sigma }{r}\right)}^{12}-{\left(\frac{\sigma }{r}\right)}^{6}\right)+\epsilon ,\;r\leq 2^{\frac{1}{6}}\sigma \\ 0,\;r>2^{\frac{1}{6}\;}\sigma \end{array}$$

The hydrodynamic volume fractions of interest range from 5% to 30%; at higher volume fractions, steric interactions between the colloids start to dominate and frustrated systems can occur [83]. All quantities are described in dimensionless units with mass in units of $m_{s}$ (the mass of the solvent particles), energy in units of ϵ, length in units of σ, force in units of $\frac{\epsilon }{\sigma }$, and time in units of $\tau =\sqrt{m_{s}\sigma^{2}/\epsilon }$.

Figure 1b and Figure 1c shows a representative snapshot of the CGMD–SRD hybrid model used for the simulations with CGMD active colloids and without active colloids in SRD solvent, respectively. In these snapshots, solvent beads are colored based on their relative velocities to illustrate their distribution. The SRD beads effectively capture both long- and short-range hydrodynamic behaviors, while the CGMD particles serve as models for colloidal micron-sized active particles (micromotors). The CGMD active particles are finite-sized spheres with a radius of 1σ and a density of $1m_{s}/\sigma^{3}.$ The simulation box spans 256 σ×256 σ in the x–y plane and 10σ in the z dimension. The confinement size was specifically chosen, designed to have an energy minimum width equivalent to that of a single CGMD bead. This selection allows beads to pass by one another, preventing geometric frustration. Simultaneously, the confinement force ensures that there is no appreciable cluster size in the *z* direction; thus, the 2D *x*–*y* spectral analysis is directly indicative of the size of the cluster.

A harmonic restraining force, applied perpendicular to the z direction, acts solely on the CGMD colloidal particles. The simulation box is periodic in all dimensions. To maintain a quasi-2D setup with a geometric confinement environment, a collision force between CGMD active particles and the system wall is implemented. This ensures that the CGMD active particles move within a plane of approximate thickness  3σ at the center, discouraging particle stacking in the z dimension while permitting particles to cross over and pass each other.

Initially, all CGMD active particles are regularly placed on a square lattice, as illustrated in Appendix A, with an areal density or particle coverage ratio (the 2D projection area coverage ratio of the colloidal particles to the 2D size of the simulation box) denoted as θ=30%. In the simulation, θ=5%, 10%, 15%, 20%, 25%, and 30%, each value corresponding to the number of active particles N=1089, 2116, 3136, 4225, 5184, and 6241, respectively. Upon particle creation, each particle has its orientation randomly set, initiating the driving force to act in a random direction at the beginning of the simulation. The driving force applied to the active particles is consistently directed along the axis of each particle. Due to the reorientation of spherical particles in a fluid, there is no net directional movement resulting from the driving force. Each active particle imparts a constant driving force F through its geometric center, with F=0, 0.5, 1, 5, and 10. As a specific case study, Appendix A demonstrates the clustering of particles over time and depicts their nearest neighbors.

## 3. Results and Discussion

### 3.1. Dynamic Clustering Behavior

The fundamental quantities of interest for this system are the number and distribution of clusters, which change over time due to the diffusion-limited aggregation of individual particles and clusters. In the context of this confined setup with a predetermined number of particles, the system tends to converge towards fewer and larger clusters. Individual particles aggregate to form clusters, and as time progresses, these clusters undergo further aggregation.

#### 3.1.1. Time Evolution of Cluster Number and Cluster Size

Figure 2 gives a visual representation of how both time and driving force affect the clustering behavior for a given coverage (in this case, θ=15%). An increase in the active driving force causes more rapid clustering behavior, especially in the initial stages of the simulation. This leads to quicker formation and aggregation of clusters compared to the passive colloidal particles (those with no driving force; F=0). The dynamics of clustering are intrinsically changed by the addition of a driving force, exhibiting more complex effects than a simple acceleration of the observed clustering. As evidenced in Figure 3, both coverage and driving force significantly contribute to the time evolution of clustering behavior, with snapshot times at t=1000τ. Higher coverages promote faster clustering, particularly at smaller times, due to the decreased average inter-bead distances resulting from the geometrical realities of increased particle coverage. As particle aggregation into clusters proceeds, the cluster size becomes a crucial metric used in conjunction with total cluster number to determine the uniformity of clustering behavior. For this section, cluster size is calculated as the total number of particles within each cluster.

Figure 4 illustrates the cluster size distribution $P\left(\bar{S}\right)$ and the mean cluster size $\bar{S}$, where $\bar{S}$ represents the number of particles within an individual cluster. Fitting lines are provided to determine the time evolution of $\bar{S}$ and the peak of the probability distribution, demonstrating how the exponents can be used to scale the cluster size distributions to a single curve. The figure shows that, for a given section of the curve, clustering behavior can be effectively represented by a simple power law. This form of scaling law for diffusion-limited cluster aggregation and growth has been shown to fit quite well with findings in previous works in the literature [84,85]. This power law allows for the documentation of not only instantaneous but also dynamic behavior in such a clustering process, providing a comprehensive characterization for future reference. This scaling is based upon the stochastic nature of clustering, which tends to smooth out for large sample sizes. Appendix A further demonstrates power-law behavior and collapse to a single curve for θ=20%, F=0.5, 465τ<t<6000τ. The lower number of clusters in this case may result in more obvious scattering from the fit observed at lower time intervals.

The probability distribution of the cluster size can be fitted well by the gamma distribution:(8)$$P\left(S\right)=\frac{b^{-a}S^{a-1}e^{-\frac{S}{b}}}{\Gamma \left(a\right)}$$

The gamma distribution has a shape parameter (a>0), a scale parameter (b>0), a cluster size (S>0), and Γ(a) is the gamma function. As shape parameter (a) increases, the distribution tends to approach a normal distribution. The gamma distribution is useful in applications where there is a physical lower bound but no non-statistical upper bound. Appendix A demonstrates a fitting of the gamma distribution to the cluster size over time for various θ and F. The fitting parameters, a and b, can be plotted to visualize the skewness and mean of the distribution. Specifically, the mean of the distribution $\bar{S}$ is calculated as a∗b, and the skewness is inversely proportional to the shape, $2/\sqrt{a}$. The plot of the scale parameter, closely linked with cluster size, reveals a flattening or reduced clustering in the middle region, as shown in Appendix A. Additionally, the mean cluster size is observed to increase as a power function of elapsed time, following the form $\bar{S}\propto t^{m}$, where m is dependent on coverage and force.

#### 3.1.2. Cluster Dynamic Spatial Distribution

Another aspect of clustering behavior, along with cluster size and number, is the physical distribution of the clusters in space [86]. The periodic system can be analyzed through fast Fourier transformation (FFT) of snapshots captured in the x–y plane, as the quasi-2D nature of this system facilitates spatial analysis. Appendix A illustrates a representative FFT transformation of a clustering image, including the circularly averaged power profile for θ=15%, F=0, t=5000τ. For each snapshot used for analysis, an image is generated as seen in Appendix A. Images are composed of 1024×1024 pixels for the 256  × 256 μm system (16 pixels/μm^2^ resolution) and generated as completely black and white. Appendix A shows a representative 2D fast Fourier transform (FFT) of a clustering image where white pixels (background) are treated as zeros, while black pixels (colloidal particles) are treated as ones. The circular average of the power of the FFT transform is calculated, and the k-space power spectra are plotted as seen in Appendix A. To find the representative length $k_{0}$, a Gaussian curve was fitted to the first peak for every snapshot. The values of $k_{0}$ and $PSD\left(k_{0}\right)$ were determined and plotted, as shown in Figure 4c,d. Figure 5 demonstrates that, for each given dynamic region, a fitting power law allows for scaling of the FFT spectra by the power-law exponent. This scaling allows all snapshot spectra over the given region to collapse onto a single curve after scaling. More information on the FFT image analysis is provided in the Appendix A.

The pre-clustering, individually separated particles prior to clustering were not suitable for FFT calculation, as demonstrated by Appendix A. In this figure, the initial square-lattice geometrical configuration of particles at setup causes crystalline FFT spectra, and the crystalline peaks gradually disappear as the system evolves to amorphous clustering. Notably, there is little to no clustering happening during this period, and consequently, no scaling laws for cluster distribution or spacing are presented or discussed for the pre-clustering behavior in this work. Figure 3 shows representative snapshots of low, medium, and high coverage and force to demonstrate how the snapshots used for FFT analysis change as a function of these parameters.

Appendix A shows the same cases as Figure 3, but at a timestep of 5000τ. Over this duration, clustering proceeds towards fewer, larger clusters across each of the cases. However, the spatial distribution of clusters remains relatively uniform, as observed consistently across the five independent runs for every variation of F and θ. Due to this spatial uniformity, the FFT analysis used can reasonably be relied upon to determine the average intercluster spacing distance, as portrayed by Appendix A. This intercluster spacing distance is represented by its inverse, $k_{0}$. It can be seen that higher coverage and driving force both accelerate the rate of clustering. At higher θ, there are more particles in solution and, thus, the rate of collision and clustering is increased. For higher F, the effective diffusion constant for the colloidal particles is higher and, thus, clustering proceeds following a scaling law with a higher exponent. Both FFT spatial distribution and clustering number can be well represented by their power-law scaling exponents.

#### 3.1.3. Phase Diagram of Dynamic Characteristics

This section presents definitions and descriptions of the different dynamic regions observed during clustering, considering the evolution of the number of clusters, cluster size distribution, as well as the spatial analyses which are discussed in detail in the preceding sections. Figure 6 shows the power-law curves fitting the different dynamic regions, and their intersection is termed as the critical time for the distinction of regions. As seen in Figure 6, there are only two distinct dynamic regions for θ=5% and 10%, but for θ=15–30%, four distinct dynamic regions occur. Table 1 briefly describes the different dynamic regions, with a detailed description as follows: The first dynamic region, Region I, is pre-clustering, where the average cluster size is approximately one (monomer region). As the coverage density grows higher, the time for this region becomes shorter due to the decrease in inter-bead spacing in the initial configuration. The length of Region I also decreases over time with an increase in driving force F, due to the increased average velocity of the active particles. In Region I, virtually no clustering can be seen; this pre-clustering region can be considered as a preliminary stage where the average cluster size is effectively zero. This region is visible in Figure 4a at t<100 τ or in the initial stages of Figure 7c. Region I becomes vanishingly small as the coverage or the driving force increases, due to the accelerated initiation of clustering at higher particle densities and velocities (Figure 7).

Region II can be considered as the initial clustering behavior, and this region dominates for θ=5–10%. As seen in Figure 7d, the driving force F has minimal effect on the clustering behavior of this region at higher coverage densities. Region III demonstrates a marked slowing of clustering, especially prominent at higher θ and lower F. This region is characterized by larger clusters and a near-complete disappearance of the monomer phase, as seen in the 500 τ snapshots given in Figure 2. At exceptionally large driving forces, this region becomes far less prominent. The final region is Region IV, representing the long-time clustering behavior. The clustering here is comparable to that observed in Region II. Clustering in this region is characterized by large clusters flocculating towards the limit of one single cluster within the system. Figure 8 displays a 3D representation of the different dynamic regions, highlighting the critical time separating them as a function of both driving force F and coverage θ. It can be seen that the transition between Region I and Region II is smooth and monotonic across all F and θ, while the transition between Region III and Region IV exhibits greater variation. Table 2 offers a detailed breakdown of the critical times for transitions between dynamic regions. Appendix A demonstrates that, for each given dynamic region (represented by Region IV, θ=15% and F=0), a fitting power law allows for scaling of the FFT spectra by the power-law exponent, which allows for all snapshot FFT spectra over the given region to collapse to a single curve. Appendix A show the representative clustering behavior evolution over time for each coverage θ and driving force F, providing a comprehensive overview of the procession of the clustering process.

### 3.2. Scaling Laws and Relationships

#### 3.2.1. Particle Coverage-Governed Scaling

One of the most important parameters influencing the clustering of colloidal particles is their packing density, represented in this work by the coverage, θ. The distinct behaviors observed in Figure 7 at low and high coverages are primarily influenced by the increased particle proximity at the beginning of the simulation and the overall larger and more numerous clusters as time proceeds. Herein, we leverage the qualitative behaviors discussed in the previous section and use the fitting exponents to quantitatively describe and scale the behavior. For each given dynamic region, the power-law fitting exponent is given in Table 3. Figure 7 demonstrates different clustering evolution behaviors for various F and θ. As can be seen, driving force F tends to flatten and accelerate the clustering behavior, in line with previous reports of driving force causing an increased effective diffusion coefficient for active particles. The clustering coverage θ, however, has a pronounced effect on the shape of the cluster number curve, with simple clustering behavior observed for low coverage (θ=5%, 10%) and a more complex time evolution of the cluster number for higher coverage (θ>10%).

#### 3.2.2. Driving Force-Governed Scaling

Appendix A shows all of the five independent runs for θ=5% at different driving forces. The similarity between cases remains consistent across different θ and F, while at F=10, occasional temporary detachment of a bead from a cluster is observed due to the similar magnitude of the driving force and inter-bead attractive potential. From Appendix A, we can see that the average cluster number and cluster size can be taken as a good approximation for all runs across each variation of driving force F for a given coverage density θ. Appendix A shows that only θ matters in terms of cluster number over time, and absolute bead number is not significant. For systems with 16x more or fewer beads, there was no big difference in clustering behavior over time, although there were minor size effects towards the end of the run when the number of beads was small ($N_{0}<500$). This can be attributed to the discrete behavior of very few (N<10) clusters when compared with the more continuous behavior of many (N≥100) clusters. Appendix A shows the absolute cluster number over time for various θ and F (rather than the normalized number shown in Figure 7). This visualization helps to highlight the rapid clustering behavior seen at higher θ and also the similarities in long-time behavior (t>3000 τ).

Figure 9 compiles different scaling exponents for the cluster size distribution and 2D cluster spatial distribution power laws across all coverages and driving forces for Region II. Within a given region, clear variations in clustering behavior as a function of θ are observed, with more pronounced differences in Region III and Region IV. Figure 10 demonstrates the differences between cluster size scaling exponents for Region II. The scaling exponent for Region II is not heavily affected by the coverage percentage, suggesting that early clustering behavior can be attributed to diffusion-driven or enhanced diffusion for F>0, with less dependence on the density of colloidal particles or micromotors. Figure 11 illustrates the differences between cluster scaling exponents ($m_{k})$ and cluster spatial distribution scaling exponents ($h_{k}$) for Regions II, III, and IV for θ=15–30%. As θ becomes larger, the scaling exponent for Region III decreases, indicating a marked slowing of the clustering behavior. However, as F increases, the scaling exponent for Region III increases rapidly, reaching a near-constant value for F=10 (near 0.8 as seen in Table 3). The different behaviors observed at low and high driving forces highlight the distinction between passive colloidal clustering and the clustering of active colloidal micromotors. Active motion not only increases the effective diffusion coefficient of Brownian particles but influences the scaling of clustering times as well [87,88,89].

## 4. Conclusions

In summary, we have established a hybrid coarse-grained molecular dynamics simulation model to investigate the dynamic clustering behavior of active particles under confinement. Several scaling laws were identified, capturing both cluster behavior and overall morphological changes. For the cluster behavior, the average cluster size $\bar{S}$ was found to scale with time (t) as $\bar{S}\propto t^{m}$ (0<m<1), and the amplitude of the cluster distribution probability $P\left(\bar{S}\right)$ followed $P\left(\bar{S}\right)\propto t^{h}$ (−1<h<0). For the morphological behavior, the representative wavevector $k_{0}$, characterizing the quasi-periodic spatial distribution of clusters, scaled as $k_{0}\propto t^{m_{k}}$ ($-1<m_{k}<0)$, and the power spectral density value at $k_{0}$, $PSD\left(k_{0}\right)$, obeyed $PSD\left(k_{0}\right)\propto t^{h_{k}}$($0<h_{k}<1).$ Detailed study shows that, depending on the colloidal particle coverage *θ*, the dynamic clustering process can be divided into four different regions: Region I (pre-clustering), Region II (initial clustering), Region III (initial cluster–cluster merging), and Region IV (final cluster–cluster merging). The behaviors in these regions were shown to strongly depend on *θ* and *F*: at low *θ* and *F*, diffusion-induced aggregation dominates the clustering dynamics; and cluster–cluster merging plays the most important role at high *θ*. The role of *F* was primarily observed in shortening the periods of Regions I and II and accelerating the cluster–cluster merging process. Due to limitations of computational resources, it is determined that a 2D system would provide valuable insights into clustering, as may occur on films or liquid surfaces or two liquid interfaces, at a larger scale than would be possible with a 3D system. However, future experiments may pursue whether the scaling laws observed in this work also present in cluster formation in 3D systems.

The study of active particles is irrevocably tied to the future of various scientific fields, especially those pertaining to medicine or health. Active particles, such as micromotors, hold promise in performing controllable tasks at the microscale, akin to natural bacteria or cells. As the field of active matter advances, active particles may become integral in scientific research work and potentially find applications in our daily lives. This work contributes to the understanding of the aggregation and possible removal of micromotors in medical and waste removal fields, particularly within laminar blood flow at biologically relevant Reynolds numbers.

## Figures and Tables

**Figure 1 nanomaterials-14-00144-f001:**
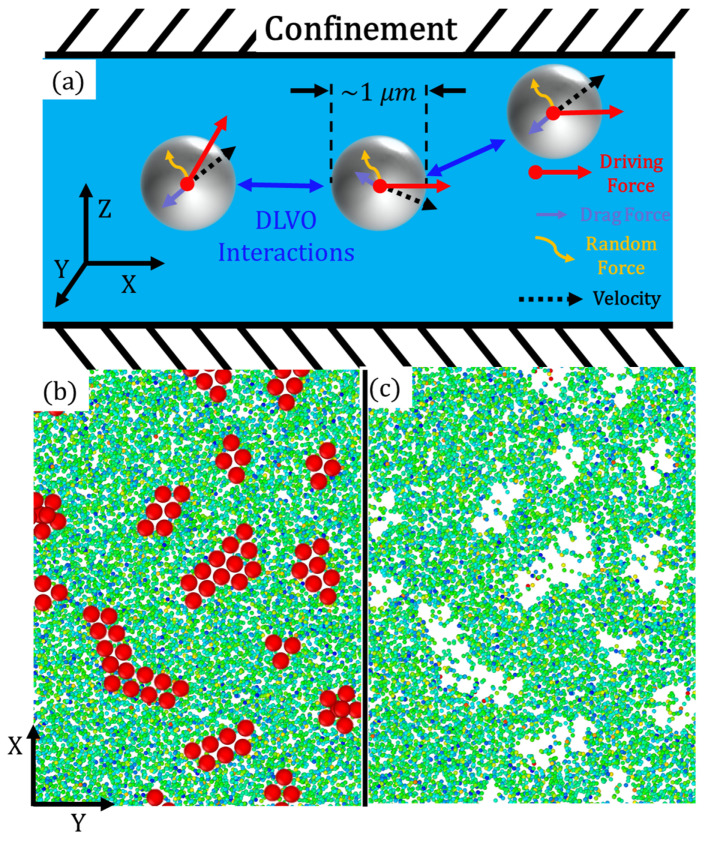
(**a**) Schematic of the spherical active particles, with on-axis driving force, under confinement. The particles can attract each other (cluster) at close distances due to the DLVO interactions. The explicit SRD fluid grants hydrodynamic effects such as drag forces in the opposite direction of motion and random thermal forces. Model setup: (**b**) The CGMD beads (red) in SRD solvent, and (**c**) the solvent with transparent beads to show solvent density. Solvent beads are colored as a function of their relative velocity, to demonstrate the distribution.

**Figure 2 nanomaterials-14-00144-f002:**
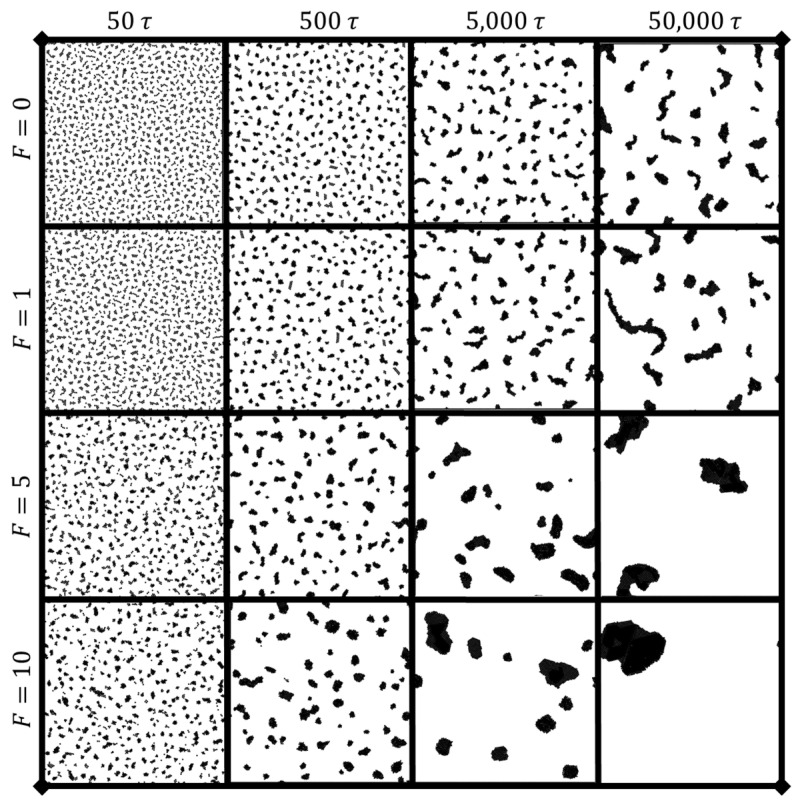
Clustering behavior over time (50 τ to 50,000 τ) for θ=15%, for different applied force F.

**Figure 3 nanomaterials-14-00144-f003:**
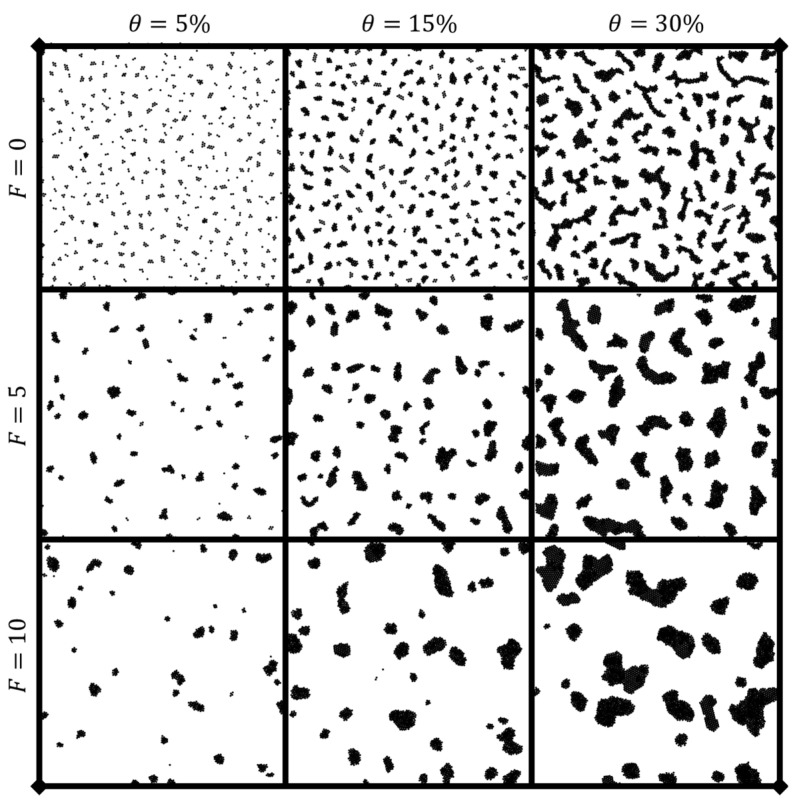
Clustering behaviors at different densities and driving forces. Time is t=1000 τ.

**Figure 4 nanomaterials-14-00144-f004:**
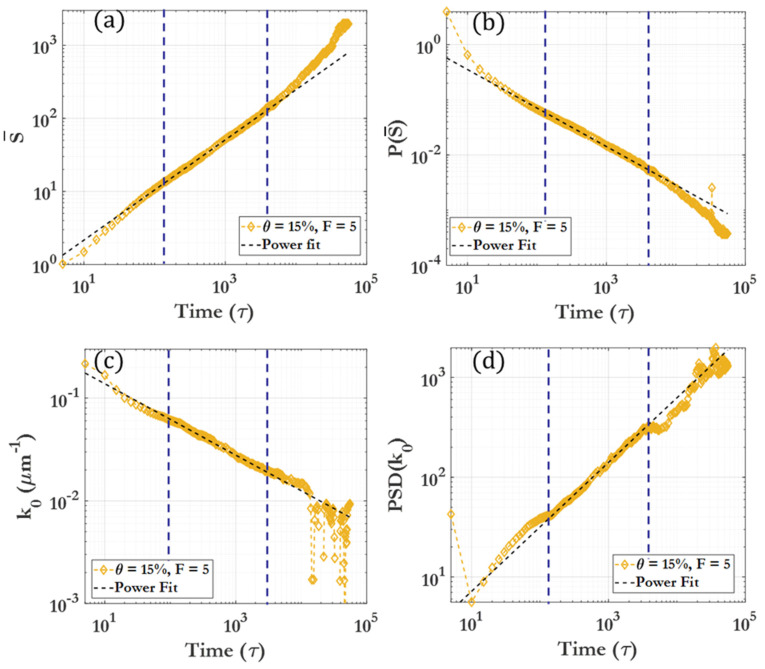
Cluster size distributions over time for θ=15%, F=5. (**a**) The mean of the cluster number distribution, with fitting line for the region t=150 τ to t=3000 τ; (**b**) the peak height of the cluster number distribution, with fitting line. (**c**) The circularly averaged FFT peak center for k_0_, with fitting line for the region of interest; (**d**) the peak intensity of k_0_, with fitting line for the region of interest. Dotted lines show the beginning and end of the region of interest.

**Figure 5 nanomaterials-14-00144-f005:**
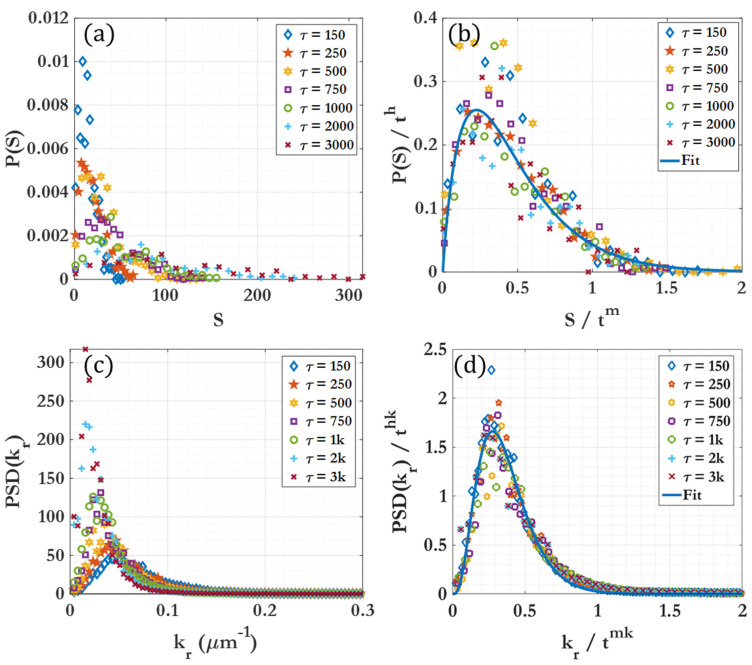
Cluster dynamics over time for θ=15%, F=5. (**a**) The cluster size distributions for t=150 τ to t=3000 τ. (**b**) The same cluster size distribution after being scaled by the scaling function for the pertinent region. (**c**) The circularly averaged FFT spectra for t=150 τ to t=3000 τ. (**d**) The same spectra after being scaled by the scaling function for the pertinent region.

**Figure 6 nanomaterials-14-00144-f006:**
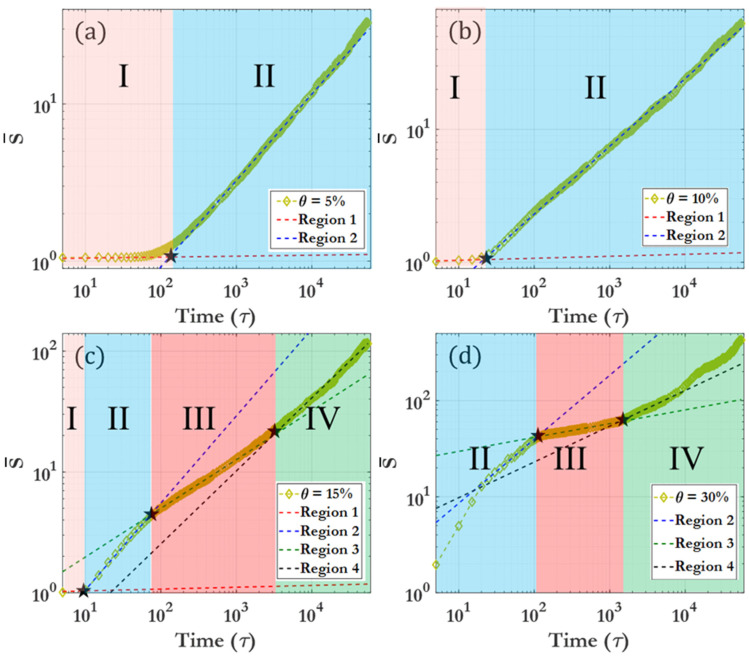
Demonstration of fitting curves for the average cluster size $\bar{S}$ and the associated different dynamic regions for (**a**) θ=5%, (**b**) θ=10%, (**c**) θ=15%, and (**d**) θ=30%. F=0 for all cases shown. The intersection points of fitting curves are denoted by asterisks.

**Figure 7 nanomaterials-14-00144-f007:**
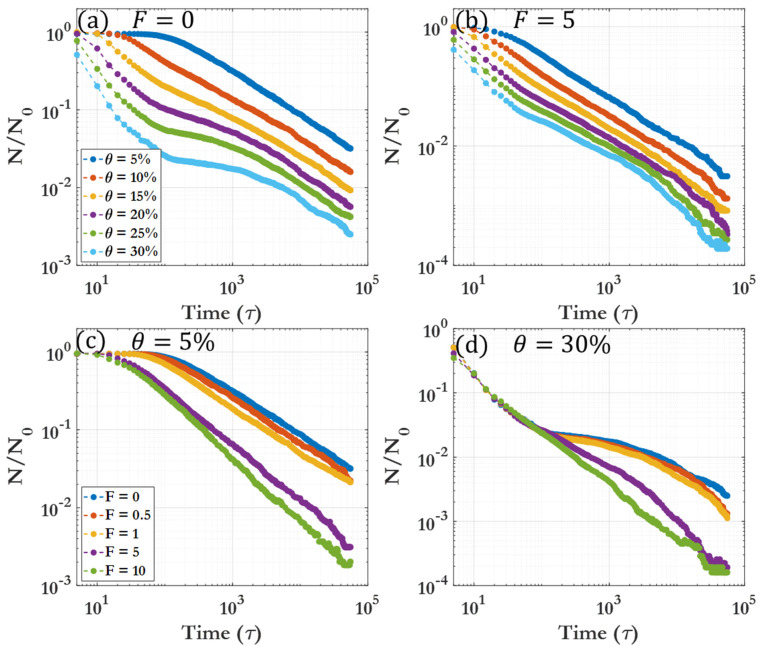
Normalized cluster number. (**a**) F=0, (**b**) F=5, (**c**) θ=5%, (**d**) θ=30%. Each case is an average of five runs. N is number of beads.

**Figure 8 nanomaterials-14-00144-f008:**
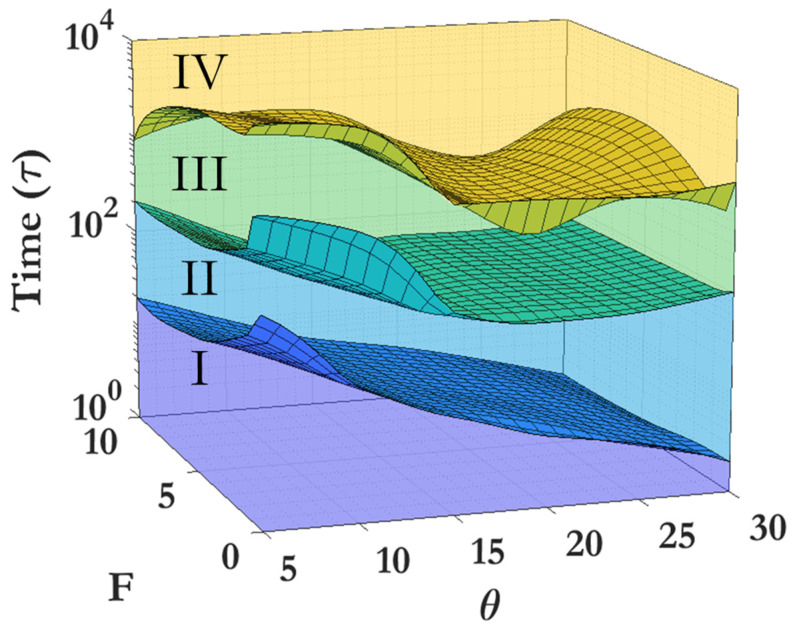
Regions I, II, III, and IV (plotted as filled colored volumes), and the transition times between them (plotted as gridded surfaces), as a function of both coverage (θ) and driving force (F). The surfaces have been smoothed between discrete points using a modified cubic spline interpolation.

**Figure 9 nanomaterials-14-00144-f009:**
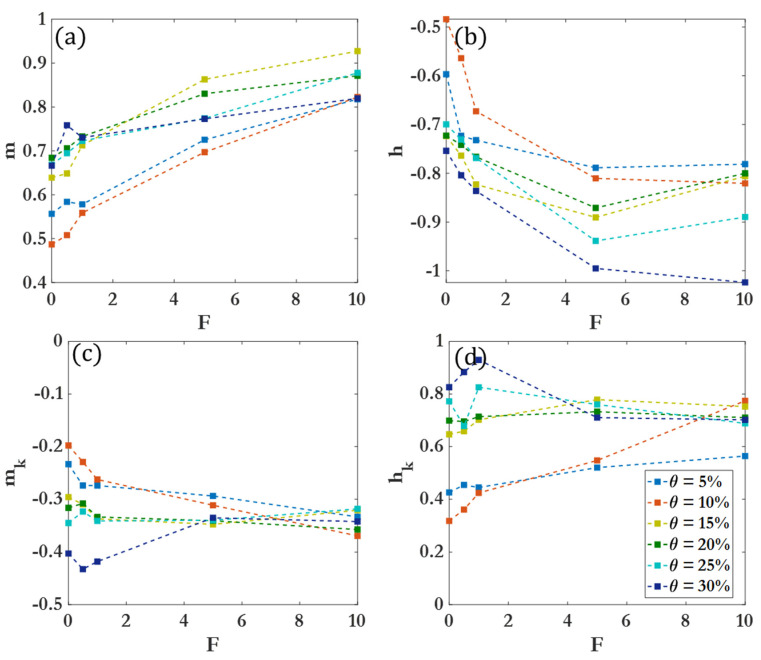
Scaling exponents (**a**) m, (**b**) h, (**c**) mk, and (**d**) hk for Region II across all θ and F tested.

**Figure 10 nanomaterials-14-00144-f010:**
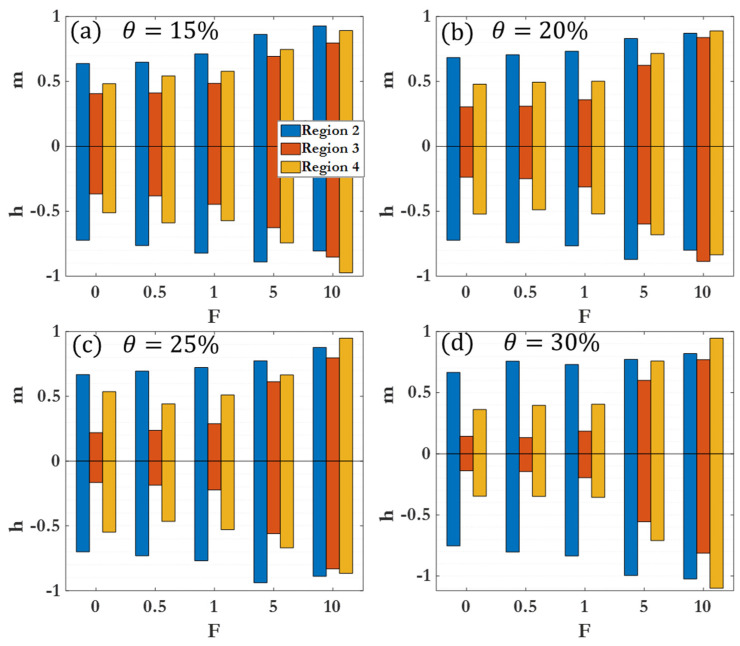
Scaling exponents m (positive) and h (negative) for different dynamic regions. (**a**) Results for θ=15% at different F; (**b**) results for θ=20% at different F; (**c**) results for θ=25% at different F; (**d**) results for θ=30% at different F.

**Figure 11 nanomaterials-14-00144-f011:**
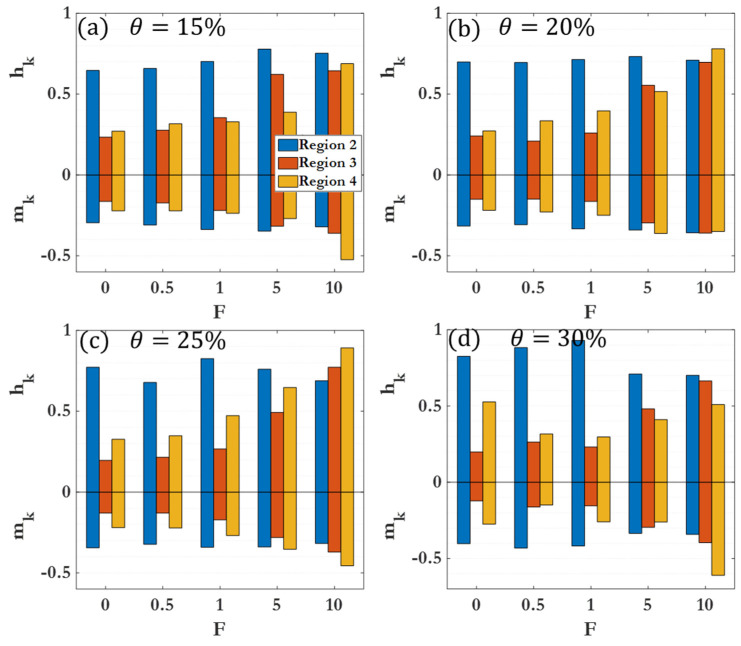
Scaling exponents $m_{k}$ (negative) and $h_{k}$ (positive) for different dynamic regions. (**a**) Results for θ=15% at different F; (**b**) results for θ=20% at different F; (**c**) results for θ=25% at different F; (**d**) results for θ=30% at different F.

**Table 1 nanomaterials-14-00144-t001:** Different dynamic regions.

**Region 1**	Effectively no clustering seen	All θ, F
**Region 2**	Initial rapid clustering	θ=15%–30%
**Region 3**	Marked slowing of clustering	θ=15%–30%
**Region 4**	Long-time clustering behavior	θ=15%–30%

**Table 2 nanomaterials-14-00144-t002:** Critical time for region transitions (t_1_, t_2_, t_3_). Units are in τ.

	F=0	F=0.5	F=1	F=5	F=10
θ=5%	129.4	93.82	65.05	23.88	19.52
θ=10%	22.05	20.71	18.42	11.83	9.78
θ=15%	9.49	9.17	8.72	6.44	5.21
	99.93	84.58	75.60	55.16	44.21
	1114	1186	1207	1180	1519
θ=20%	5.19	5.09	5.56	4.22	3.61
	69.41	65.74	70.78	67.26	60.06
	1094	464.3	497.2	501.7	636.7
θ=25%	3.66	3.15	3.11	2.88	2.25
	75.41	72.80	72.29	64.64	59.54
	1014	901.6	803.6	687.7	383.2
θ=30%	2.01	1.99	1.98	1.49	1.31
	102.2	91.96	88.46	80.81	73.54
	1234	589.4	623.5	1813	1013

**Table 3 nanomaterials-14-00144-t003:** Dynamic region scaling exponent m for Regions II–IV.

	F=0	F=0.5	F=1	F=5	F=10
θ=5%	0.5454	0.5837	0.5781	0.7252	0.8177
θ=10%	0.4867	0.5078	0.5587	0.6969	0.8226
θ=15%	0.6388	0.6484	0.7126	0.8629	0.9273
	0.4065	0.4115	0.4859	0.6931	0.7962
	0.4824	0.5433	0.5779	0.7468	0.8922
θ=20%	0.6840	0.7057	0.7327	0.8302	0.8711
	0.3039	0.3100	0.3595	0.6246	0.8389
	0.4793	0.4939	0.5018	0.7161	0.8896
θ=25%	0.6682	0.6946	0.7227	0.7740	0.8778
	0.2197	0.2379	0.2886	0.6132	0.7971
	0.5359	0.4422	0.5107	0.6647	0.9489
θ=30%	0.6660	0.7582	0.7302	0.7729	0.8195
	0.1432	0.1323	0.1848	0.6006	0.7693
	0.3621	0.3955	0.4053	0.7583	0.9453

## Data Availability

Data are contained within the article.

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
