# Peer review of "Dynamic Clustering and Scaling Behavior of Active Particles under Confinement"

_nanomaterials, 2024, doi:10.3390/nano14020144_

Round 1

Reviewer 1 Report

Comments and Suggestions for Authors

In this computational work, the authors used a hybrid coarse-grained molecular dynamics-stochastic rotation dynamics (CGMD-SRD) method to investigate the dynamic clustering behavior of active particles (influenced by an external field) under geometric confinement. Overall, the topic is of general interest to the nanomaterials community, the work is novel, the manuscript is well-written and organized, and the authors have done a great job of describing their methodology and results. The work is technically sound, and the conclusions are supported by a comprehensive set of computational data and their analyses. I recommend publication in Nanomaterials pending some minor revisions to the manuscript to correct/clarify the following points:

1.     Line 205: Please correct “CDMD” to “CGMD.”

2.   Table 2 (Page 15) and Table 3 (Page 16): The column headers are missing in the tables. I presume each column represents a given driving force (F) or maybe runs? Please clarify and correct.

3.   It would be interesting to compare the dynamic clustering behavior of the active particles under confinement to their behavior in an unbound system under the same coverage and driving force conditions considered in this work. Would the authors be able to comment on the differences? I understand this may not be within the scope of this work, but some statements would be useful to the reader. Also, along the same line of thought, please provide some clarification on the confinement size in this work and its effect on the clustering behavior of the active particles.

4. As a mere suggestion, would the authors be able to validate their scaling relationships with any experimental data obtained for comparable colloidal systems under confinement? Doing so would significantly increase the applied aspects of the work.

Author Response

Please see attached response letter.

Reviewer 2 Report

Comments and Suggestions for Authors

The manuscript reports an interesting simulation study of the aggregation behavior of active particles under confinement carried out with coarse-grained molecular dynamics simulations. The results are of interest, so that the manuscript should be published with only some minor changes, basically aimed to improve its clarity and readability. My comments, observations and suggestions are listed below.

1. The numerical values of the parameters in Eqs. (4)-(7) should be explicitly given, unless they are adsorbed in the adopted dimensionless units.

2. In Fig. 4, panels c and d report the peak center and intensity (or height?) of the FFT discussed later in a different Section (see at lines 303 and 304 of Sect. 3.1.2). Accordingly, the results shown in these panels are puzzling, here.

3. As far as I can understand, in Table 2 t1, t2 and t3 correspond to the different rows for each theta value, whenever relevant. But what do the different columns stand for? Possibly for different F values? But only 4 different F values were considered, I feel. Please clarify this issue. The same applies also to Table 3.

4. One wonders whether it is possible to extract some physical information from the critical exponents in the Conclusions, for instance about their sign or their (absolute) value which is always found to be less than 1.

Author Response

Please see attached response letter.
